# Endothelial Dysfunction in Systemic Sclerosis

**DOI:** 10.3390/ijms241814385

**Published:** 2023-09-21

**Authors:** Eshaan Patnaik, Matthew Lyons, Kimberly Tran, Debendra Pattanaik

**Affiliations:** 1Department of Biology, Memphis University School, Memphis, TN 38119, USA; eshaan.patnaik@musowls.org; 2Division of Rheumatology, University of Tennessee Health Sciences Center, Memphis, TN 38163, USA; mlyons8@uthsc.edu (M.L.); ktran17@uthsc.edu (K.T.)

**Keywords:** systemic sclerosis, endothelial cell, angiogenesis, vasculogenesis

## Abstract

Systemic sclerosis, commonly known as scleroderma, is an autoimmune disorder characterized by vascular abnormalities, autoimmunity, and multiorgan fibrosis. The exact etiology is not known but believed to be triggered by environmental agents in a genetically susceptible host. Vascular symptoms such as the Raynaud phenomenon often precede other fibrotic manifestations such as skin thickening indicating that vascular dysfunction is the primary event. Endothelial damage and activation occur early, possibly triggered by various infectious agents and autoantibodies. Endothelial dysfunction, along with defects in endothelial progenitor cells, leads to defective angiogenesis and vasculogenesis. Endothelial to mesenchymal cell transformation is another seminal event during pathogenesis that progresses to tissue fibrosis. The goal of the review is to discuss the molecular aspect of the endothelial dysfunction that leads to the development of systemic sclerosis.

## 1. Introduction

Systemic sclerosis (SSc) is an autoimmune connective tissue disease. The disease is characterized by vascular defects, progressive fibrosis, and skin thickening [1]. There are two known types of SSc: limited and diffuse. These two forms can be distinguished by variations in the degree of skin involvement, association with specific autoantibodies, and the pattern of organ involvement. While the exact cause of SSc is unknown, the disease is believed to be a result of environmental factors and genetic predisposition. Family history plays a role in the development of SSc, and it can amplify the risk of developing the disease in an individual [1]. Three distinct underlying mechanism drive the disease: (a) aberrant innate and adaptive immunity leading to production of autoantibodies and cellular autoimmunity; (b) vascular dysfunction, and (c); defective fibroblasts leading to excessive collagen and matrix component deposition in vessels and various organs including skin [1]. 

Impaired vascular tone and permeability are some of the earliest signs of vascular dysfunction [2]. Dysregulation of vascular tone leading to vasospasm and compromised blood flow is the underlying mechanism of vascular dysfunction. This dysregulation of vascular tone results from the imbalances of vasodilator mediators, e.g., nitric oxide (NO) and vasoconstrictor, e.g., endothelin. Besides this, cellular changes such as large gaps between endothelial cells, vacuolization of endothelial cell cytoplasm, loss of membrane bound storage vesicles are some of the earliest endothelial cell changes [3,4,5]. Along with the endothelial cell changes, capillary enlargement and capillary losses continue progressively and furthermore intimal proliferation and proteoglycans accumulate in arterioles and small arteries [6,7].

The vascular endothelium maintains vascular tone, regulates platelet function, and controls inflammation. It plays a significant role in the tissue remodeling through angiogenesis and vasculogenesis [8]. Early in the disease process, typical vascular symptoms, e.g., Raynaud’s phenomenon precede tissue fibrosis suggesting that microvascular injuries and dysfunctions involving the vascular endothelial cells (VECs) and capillaries are the initial event [9]. This clinical observation led to the conclusion, as early as 1975, that the primary cause of SSc involves endothelial damage and dysfunction [10]. Furthermore, the interaction of endothelial cells with other cells, e.g., smooth muscles, platelets, fibroblasts, and pathways, e.g., the coagulation system and immune system drive the disease process [9]. 

Four major distinct processes involving the endothelial cells are crucial in the pathogenesis of SSc: Endothelial cell injury, defective angiogenesis, defective vasculogenesis, and endothelial to mesenchymal transformation (Figure 1). These processes are discussed more in detail below.

## 2. Endothelial Cell Injury

Initial vascular endothelial cell damage followed by vascular remodeling with arteriole intimal proliferation, capillary breakdown and vascular occlusion are some of the key events in the progression of SSc [11]. Endothelial cell injury is accompanied by lymphocytic and mononuclear lymphocytic perivascular infiltration in the affected tissues are few of the earliest changes seen in SSc [12]. Infectious agents, cytotoxic T cells, autoantibodies against the endothelial cells and NO-related free radicals have been cited for the endothelial cell damage but the exact role of each of these agents is not clear [2]. Evidence of endothelial cell injury, e.g., high serum levels of circulating von Willebrand (VW) factor, endothelin-1, circulating viable and dead endothelial cells, and soluble JAM-1 are present in the circulation [2,13,14,15,16,17]. There is increased expression of the genes encoding the von Willebrand factor (vWF), tissue factor (F3), ephrin A1, and ET-1 in stimulated endothelial cells suggestive of endothelial cell activation and damage [18]. Among the infectious agents, viral agents have been proposed as possible trigger of endothelial cell injury [19]. Cytomegalovirus has been implicated in the pathogenesis of SSc for its potential to infect endothelial cells and there is increased prevalence of anti-Hcmv antibodies in the sera of subjects with SSc [20,21]. There could also be activation of the autoreactive B cell clones through molecular mimicry as anti-topoisomerase I antibodies recognize a pentapeptide of the autoantigen-sharing homology with the hCMV-derived UL70 protein [22]. There is presence of anti-HCMV antibodies directed against an epitope (VTL GGAGIWLPP) contained within UL94, a human cytomegalovirus-derived protein expressed in the infected endothelial cells. UL94 is localized in the nucleus of infected endothelial cells and may be responsible for the regulation of viral and/or cellular gene expression. The antibodies bind to the endothelial cell surface receptor NAG-2, bearing similarity to UL94 and cause endothelial apoptosis, which is a key event in SSc pathogenesis [18,23]. There is a significantly higher prevalence of HHV-6A/B DNA in both the blood and peripheral tissues of SSc patients compared to controls [24]. Even among the SSc patients one subgroup has higher viral load compared to another group suggesting that it may be responsible for a subgroup of patients. The authors were further able to demonstrate that human herpes virus A and B (HHV-6 A and B) interferes with normal function of the endothelial cells and causes pro-fibrotic molecule expression in the endothelial cells. Prior parvovirus B19 infection could result in increased expression of TNF-α in the endothelial cells, which may be of pathologic significance in SSc [25]. Epstein bar virus (EBV) has also been implicated in endothelial cell injury in SSc [26]. In an in vitro model, the investigators demonstrated that the human monocytes bound to the recombinant EBV act as a shuttle, EBV virus can infect the endothelial cells, and there is overexpression of the EBV early lytic genes. The EBV induces activation of the TLR9 innate immune response and type I IFN. Type I interferons (IFNs) are known to promote endothelial cell death, inhibit endothelial cell migration [27], and may play a role in endothelial cell injury in SSc [26]. It is possible that the expression of the EBV early lytic genes in the infected endothelial cells may cause vascular endothelial cell apoptosis [26]. Even though SARS-CoV-2 infection is a more acute process, it has a lot of similarity with early endothelial damage and vasculopathy seen in SSc [28]. CD4+ Cytotoxic T-cells and CD8+ cytotoxic T-cells are expanded in early SSc patients [29]. They are found near the endothelial cells undergoing apoptosis. Peripheral T-cell tolerance breakdown may lead these T-cells to interact with the endothelial cells presenting self-antigens in conjunction with HLA class II or HLA class I molecule resulting in the endothelial cell apoptosis. Apoptotic cells trigger TAM family kinases on macrophages and potentially could accelerate wound healing [30]. In another study, the authors found that CD226+CD8+ T cells produced a higher number of various cytokines than CD226^−^ ones, and CD226^high^CD8+ T cells from SSc patients showed upregulated IL-13 production and positive correlation with the cytotoxic capacity of CD8+ T cells against human umbilical vascular endothelial cells (HUVECs) [31]. However, neutralization of CD226 in CD8+ T cells impaired costimulation, cytokine production, and cytolysis against HUVECs. Ischemia reperfusion events, e.g., the Raynaud phenomenon, results in oxidative stress in SSc leading to the formation of reactive oxygen species, e.g., including superoxide anion radical (O_2_•), hydroxyl anion (•OH), and hydrogen peroxide (H_2_O_2_) [32]. The source of these reactive oxygen species varies from the peripheral blood cells in the vessel lumen or monocytes, endothelial cells, fibroblasts in response to various noxious stimuli. These reactive oxygen species inhibit the release of NO, prostacyclin, tissue plasminogen activator, protein S and heparin sulphate from the endothelial cells leading to the alteration of the vascular tone [33]. Anti endothelial cell antibodies (AECA) are found in 22–68% of patients with SSc [34]. These antibodies cause endothelial cell apoptosis by both the fas-independent mechanism and the antibody-dependent cellular cytotoxicity mechanism [35,36]. In an invitro model, in human microvascular endothelial cells (HMVECs), when cultured with AECA, there is endothelial apoptosis and the HMVECs release endothelin 1 [37]. AECAs in SSc are linkedto higher levels of total and activated circulating endothelial cells suggestive of endothelial cell damage. Levels of AECAs are correlated with the number of apoptotic endothelial microparticles. Impairment of vascular repair by having fewer endothelial cell progenitors is seen in subjects with AECAs [38]. Immune complexes (ICs) containing SSc specific antibodies (anti-centromere, anti-topoisomerase I, and anti- RNA polymerase) induce the endothelial cells to a pro-inflammatory and pro-fibrotic phenotype [39]. Incubation of the endothelial cells with SSc-ICs increased the expression of several molecules which can cause vascular dysfunction (ET-1, IL-8), inflammation (IL-6 and ICAM-1), and fibrosis (TGF-β1). 

## 3. Defective Angiogenesis

Angiogenesis is dysregulated in SSc indicating defective microvascular cell differentiation from the preexisting vessels to form new blood vessels [40,41]. The endothelial cells showed reduced angiogenic properties [42,43]. These cells have intrinsic defect in NO production due to the downregulation of the endothelial NO synthase [44]. There is overexpression of proangiogenic factors like VEGF-1 and endothelin-1 [45,46], but VEGF receptor signaling may be impaired [42]. There is also an impaired response to pro-angiogenic chemokines Gro-γ/CXCL3, GCP-2/CXCL6, or CXCL16. The endothelial cells do not migrate in response to the chemokines and vascular endothelial growth factor (VEGF) because of defective signaling pathways. Studies have found that the anti-angiogenicVEGF_165_ b isoform of VEGF is overexpressed compared to the proangiogenic VEGF_165_. Binding of VEGF_165_ to the tyrosine kinase receptor VEGFR-2 results in insufficient tyrosine kinase phosphorylation/activation and incomplete signaling, leading to incomplete angiogenic response. The authors have also demonstrated that the plasma level of VEGF_165_ b is associated with severity of microvascular damage suggesting that this isoform of VEGF is responsible for vasculopathy in SSc [47,48]. Vascular abnormality and reduced capillary density alter blood supply and tissue oxygenation in SSc. Chronic hypoxia leads to an increase in VEGF level and disturbs VEGF receptor signaling [49]. VEGF promotes angiogenesis by activating the endothelial cells and the endothelial progenitor cells [50]. VEGF further promotes angiogenesis by acting synergistically with the platelet derived growth factor (PDGF) and fibroblast growth factor [51]. Aberrant overexpression of the VEGF-like seen in SSc induces vascular malformations [52]. The vascular malformation is attributable to two specific endothelial cell activities: (i) neovascularization in normally avascular areas and (ii) the unregulated, excessive fusion of vessels. Hypoxia-inducible-factor-1-ά, though, is expected to be high in SSc because of ongoing hypoxia but studies have shown otherwise, levels are low in the skin compared to the healthy controls [53]. Another mechanism of angiogenesis inhibition is through increased glycolytic metabolism of dermal fibroblasts leading to extracellular acidification [54]. The authors have demonstrated that acidosis in general and lactic acidosis impair in vitro endothelial cell capillary network/lumen formation and invasion without altering cell viability. The abnormality was corrected following PH adjustment. This response to acidic PH is secondary to MMP-12 upregulation by endothelial cells and expression of cleaved uPAR (urokinase-type plasminogen activator receptor). Endostatin, an inhibitor of angiogenesis has been found to be upregulated in SSc [55]. There is alteration of signal transduction, e.g.,: ang/Tie signaling pathways [56] and Upar truncation [57]. Friend leukemia virus integration 1 (FLI1) is a transcription factor and its expression is suppressed both genetically and epigenetically in SSc patients [58]. CCN1 expression was suppressed uniformly and remarkably in dermal blood vessels of Fli1(+/−) mice and partially in those of endothelial cell-specific Fli1 knockout mice [59]. Serum CCN1 levels were significantly decreased in SSc patients with previous and current histories of digital ulcers, as compared to those without the ulcerations. This suggests that epigenetic endothelial CCN1 downregulation is at least partially due to Fli1 deficiency and may contribute to the development of digital ulcers in SSc patients [59]. FLI1 deficiency suppresses the expression of CD31, VE-cadherin, S1P1, and PDGF-B in the endothelial cells, while upregulating matrix metalloproteinase-9 and CCR6 resulting in vascular destabilization and angiogenesis [60]. Desmoglein-2 belongs to the family of desmosomal cadherins and is involved in cell adhesion, morphogenesis, cytoskeletal organization, and cell sorting/migration [61]. There is reduced expression of desmoglein-2 (*DSG2*) in the endothelial cells of SSc patients shown by differential transcriptome profiling and by immunohistochemistry of the endothelial cells (EC) [62]. This has been linked to defective angiogenesis in SSc [63]. Circulating CXCL4 levels are elevated in SSc [64]. The proliferation, migration, and tube formation of human umbilical vein endothelial cells (HUVECs) are inhibited by CXCL4 or SSc derived plasma. This is reversed by the CXCL4 neutralizing antibody. CXCL4 downregulates Friend leukemia integration factor-1 (Fli-1) via c-able signaling in the endothelial cells and inhibits angiogenesis [64]. 

## 4. Defective Vasculogenesis

There has been a lot of interest in the role of endothelial progenitor cells which contribute to the postnatal vasculogenesis in SSc [65]. These cells are identified in flow cytometry as CD34+, CD133+, and/or CD309/VEGFR2+ [66]. There is correlation between digital ulcers and a reduced number of endothelial progenitor cells in SSc patients [66,67,68]. Low progenitor cell numbers are independent predictors of new digital ulcers on longitudinal follow up and associated with late pattern nail fold capillaroscopic changes in SSc [69,70]. In the Fra-2 transgenic mice model, the pulmonary vasculature remodeling was attenuated by blocking MMP-10 and there is increased expression of stromelysin 2 (MMP-10) in pulmonary vasculature and serum among subjects with SSc [71]. Multiple initial studies have been conflicting regarding the number of circulating endothelial progenitor cells in subjects with SSc [66,67,68,72,73]. However, these conflicting results, later found to be from different protocols used in different labs, and now using standardized protocols, it has been found that circulating progenitor endothelial cells are in fact reduced in numbers compared to the healthy controls [74]. Circulating lymphatic endothelial progenitor cells (CD34+CD133+VEGFR3+), are also found to be reduced in SSc subjects with active digital ulcers [75]. Besides that, there is inability of endothelial progenitor cells in SSc patients to differentiate into mature endothelial cells. This has been demonstrated in in vitro cultures using angiogenic growth factors [76] and using a murine tumor neovascularization model [77]. Several mechanisms have been proposed behind the qualitative and qualitative defects of the endothelial progenitor cells. An altered bone marrow microenvironment, as evidenced by increased fibrosis and reduced microvascular density, may alter progenitor endothelial capacity to differentiate into the endothelial cells [78]. Circulating pentraxin-3 potentially inhibits progenitor cell differentiation through FGF2 mediated progenitor cell differentiation. The authors have demonstrated that the number of progenitor cell counts varies inversely with circulating levels of Pentraxin-3 [79]. It has also been found that endothelial progenitor cells are defective before they are released from the bone marrow as they do not differentiate well in long-term culture with a proangiogenic growth factor [67]. Others have postulated that progenitor cells may undergo immune mediated apoptosis in circulation, which is mediated through the Akt-FOX03a-bim signaling pathway [72]. 

## 5. Endothelial to Mesenchymal Transition (EndoMT)

In SSc, there is subendothelial accumulation of activated fibroblasts or myofibroblasts in small arterioles in the lungs and kidneys which produce fibrotic tissue [80]. It has been demonstrated that some of these mesenchymal cells have originated from the endothelial cell transformation (EndoMT) [80]. Cipriani et al. showed endothelial cells showed reduced expression of vWF, CD31, and VE-cadherin (endothelial cell markers) and upregulation sSm22, α-SMA, and collagen (markers of fibrosis) suggesting the mesenchymal transformation of endothelial cells [81]. Von Willebrand factor/α-smooth muscle actin-positive endothelial cells are found in up to 5% of pulmonary vessels in subjects with SSc-associated pulmonary arterial hypertension and in the hypoxia/SU5416 mouse model [82]. The investigators demonstrated that transformed EndoMT cells on stimulation by inflammatory cytokines, e.g., IL-1 β, TNF-α, TGF β caused actin cytoskeleton reorganization, and the induction of a mesenchymal morphology. These cells showed up-regulation of mesenchymal markers, including collagen type I and α-smooth muscle actin, and a reduction in the endothelial cell and junctional proteins, including von Willebrand factor, CD31, occludin, and vascular endothelial-cadherin. Induced EndoMT monolayers failed to form viable biological barriers and induced enhanced leak in co-culture with pulmonary artery endothelial cells [82]. CD31+/CD102+EC isolated from SSc lungs expressed simultaneously mesenchymal and EC-specific transcripts and proteins [83,84]. This suggests occurrence of EndoMT in lung tissues from patients with SSc-associated ILD. Cells in intermediate stages of EndoMT were identified in dermal vessels of either patients with SSc or bleomycin-induced and urokinase-type plasminogen activator receptor (uPAR)-deficient mouse models [85]. Various cytokines and growth factors, e.g., TGF-β, PDGF, VEGF, and ET-1 activates the vascular endothelial cell and modulates the expression of adhesion molecules on the endothelial cell surface leading to the recruitment of macrophages and B as well as T lymphocytes [86,87,88]. These inflammatory cells further release various growth factors, e.g., TGF-β and CTGF, in the tissue activating the endothelial cells to release endothelin-1, which then activates the endoMT cells [89,90]. Levels of leukotriene A4 hydrolase (LTA4 H), an enzyme for LTB4 synthesis, LTB4, and its receptor, BLT1 were increased in lesional areas of the skin and lungs of SSc patients, and were abundant in myofibroblasts and endothelial cells [91]. Fibroblast-myofibroblast and endothelial-mesenchymal transitions (EndoMT) were promoted via BLT1, and are dependent on activation of the phosphatidylinositol 3-kinase (PI3K)/Akt/mechanistic target of rapamycin (mTOR) pathway. Various cytokines and other factors are involved in the EndoMT process, including TGF-β, particularly the TGF-β 1 [92], Endothelin-1 [81], Wnt3a protein [93], IL-1β [94], TNF-α [95], IFN-γ [96], microRNAs [97], oxidative stress [98], hypoxia, and hypoxia inducible factor [99]. The growth factors, cytokines, and their mechanisms of actions are summarized in Table 1.

## 6. Other Cytokines and Endothelial Dysfunction

A higher serum level of IL-33 and abnormal expression of IL-33 are found in the skin of subjects with early stage SSc [100,101,102,103]. The IL-33–ST2 axis increases proliferation, migration, and differentiation of endothelial cells with increased permeability and angiogenesis [103]. It is possible that IL-33 might mediate early events of SSc through recruitment and stimulation of ST-2 expressing cells such as immune cells, fibroblast, and myofibroblast [104]. IL-17 produced by endothelial induces expression of adhesion molecules and chemokines in HUVEC. It mediates vascular inflammation through ERK phosphorylation [105]. 

## 7. Endothelial Cell Interactions with Other Cells

Interaction between ECs and platelets is important for regulating vascular tone. Activated platelets induce thymic stromal lymphopoietin (TSLP) in human dermal microvascular ECs, inducing profibrotic and endothelial cell activating factors such as IL-13 [11]. Endothelial cells interact with the coagulation system through upregulation of multiple markers of coagulation, such as VEGF, leading to a prothrombotic state [106]. 

## 8. Clinical Manifestations of Endothelial Dysfunction

The Raynaud phenomenon, as mentioned earlier, is one of the earliest and most frequent manifestations of SSc affecting up to 95% of patients [107] It results from the altered vascular tone secondary to the imbalance between vasodilator mediators: Nitric oxide, prostaglandin I2 (PGI2), and vasoconstrictor endothelin 1. Progressive damage to the vasculature with alteration of the vessel wall by proliferative changes leads to obliterative vasculopathy and tissue damage. Frequent episodes of vasospasm, tissue hypoxia, superoxide radicals, and altered microvasculature contribute to developing digital ulcers in up to 30% of SSc patients each year [107,108]. Another form of severe vascular manifestations in SSc includes scleroderma renal crisis (SRC) seen in up to 10% of all SSc patients [109]. It is characterized by an extreme rise in blood pressure accompanied by renal dysfunction, microangiopathic hemolytic anemia, and thrombocytopenia. Though the exact pathophysiology is poorly understood, it is believed that poor renal blood flow secondary to endothelial damage, intimal hyperplasia, and narrowing of renal arterioles play a role [109]. IL-6 has been implicated in the pathogenesis of SRC [110]. Simon et al. demonstrated that in subjects with SRC, autoantibodies against Protease-activated receptor-1 (PAR-1) induced endothelial cells to produce IL-6 with increased ERK 1/2, AKT, and p70S6K signaling, as well as increased activity of the c-FOS/AP-1 transcriptional factor [110]. The authors suggested that the blockade of the c-FOS/AP-1 pathway and endothelial PAR-1 receptor would mitigate endothelial IL-6 production. Penn et al. showed an increased level of Endothelin-1 and endothelin A and B receptor expression in renal biopsy of SRC patients, suggesting that endothelin blockade may be a therapeutic approach in SRC [111]. Another serious vascular complication in SSc is pulmonary arterial hypertension (PAH) seen in up to 15% of patients [112]. Endothelial dysfunction contributes to SSc-associated PAH, leading to the expansion and aggregation of collagen type-1 positive cells and α-SMA cells in pulmonary vasculature [82]. The authors have shown that EndoMT, though infrequent, contributes to vascular remodeling and inflammatory infiltration in pulmonary vasculature of SSc PAH patients. In SSc PAH patients, there is an increase in endothelin-1 production by endothelial cells and a reduction in the production of vasodilators such as nitric oxide and PGI2 [112]. In normal healthy subjects, endothelin-1 binds to the ET_A_ receptor which is expressed in endothelial cells causing vasoconstriction and results in vasodilation by binding to the ET_B_ receptor in smooth muscle cells. In SSc PAH patients, there is upregulation of ET_B_ receptors in muscles and downregulation in endothelial cells [112]. Endothelin 1, produced by endothelial cells, binds to the ET_B_ receptor and leads to unregulated cell proliferation, resulting in vascular occlusion and an increase in pulmonary vascular resistance [112]. Endothelial cells, present in the plexiform lesions of pulmonary vasculature of patients with idiopathic pulmonary hypertension, express IL-32, which is likely involved in the proliferation and activation of these abnormal endothelial cells [113]. IL-32 levels have been found to be significantly higher among SSc PAH subjects compared to SSc subjects without PAH and can be a marker of the presence of PAH [114]. 

## 9. Evaluation of Endothelial Dysfunction

### 9.1. Morphological Assessment

Nailfold capillaroscopy is a safe well-established method for assessing capillary microcirculation in SSc. It has been a widely used technique to investigate and monitor patients with Raynaud phenomenon. Subjects can be diagnosed as having SSc based on specific patterns of capillary architecture known as “scleroderma patterns”, which can be further divided into three patterns (early, active, and late) based on the stage of disease [115]. The early pattern is recognized as having giant capillaries with normal capillary distribution, followed by the active pattern: giant capillaries, hemorrhage and disorganized distribution and loss of capillaries, and finally the late pattern is identified as loss of capillaries and capillaries with abnormal shapes [115]. These vascular patterns are reflective of the severity and progression of the disease, from mild to severe/progressive disease [116]. 

### 9.2. Functional Assessment

Various other methods of assessing the functional aspect of microcirculation have been adopted based on the principle of producing vasodilation in response to various stimuli, e.g., physical, mechanical, and chemical etc. [117]. The vascular response is determined by both the structural and functional condition of the vasculature. The various methods are as follows: peripheral arterial tonometry, laser doppler flowmetry, laser speckle contrast imaging, laser doppler imaging, laser speckle contrast analysis, and laser doppler imaging to assess small digital vessels. Flow-mediated vasodilation of the brachial artery is used to asses medium vessels [117]. Differential vascular response to NO donators (e.g., glycerol-trinitrate) or direct non–NO donators, such as adenosine, can be used to differentiate endothelium dependent and endothelium independent response [118]. Alteration in the vascular structure and smooth muscle cell leads to impaired endothelial-independent function rather than changes in the endothelium [118]. 

## 10. Treatment Strategies for Endothelial Dysfunction

### 10.1. Calcium Channel Blockers (Dihydropyridines)

Dihydropyridines, e.g., amlodipine, nifedipine, and nicardipine bind to the L-type voltage-gated calcium channels on cells, reduce influx of extracellular calcium, and induce smooth muscle relaxation, resulting in vasodilation [119]. In vitro studies have shown the protective effect of calcium channel blockers on the endothelial cells from oxidative injury [120]. Calcium channel blockers decrease plasma markers of oxidative stress in SSc patients [121]. Calcium channel blockers have been widely used and are effective in uncomplicated Raynaud phenomenon.

### 10.2. Phosphodiaterase-5 (PDE-5) Inhibitors

PDE5 inhibitors such as sildenafil and tadalafil increase the intracellular cyclic guanosine monophosphate (c-GMP) level by inhibiting PDEs. C-GMP leads to vasodilation through vascular smooth muscle relaxation, increases apoptosis, and decreases proliferation of pulmonary artery smooth muscle cells [122]. PDE-5 inhibitors have been found to be beneficial in treatment of Raynaud phenomenon, digital ulcers, and pulmonary arterial hypertension [123]. 

### 10.3. Soluble Guanylate Cyclase Activators and Stimulators

These agents increase c-GMP production through binding to soluble guanyl cyclase (sGC). The sGC stimulators bind to reduced, heme-containing sGC independent of NO and sensitizes the sGC through stabilization of sGC–NO binding to low levels of NO. sGC activators bind to the oxidized heme-free and NO-unresponsive form of sGC [124]. These agents have an anti-inflammatory, antifibrotic, and antihypertensive effect [124]. In a in vitro model, a sGC stimulator, MK-2947, increased the migratory and proliferative effect of SSc-microvascular endothelial cells and inhibited the endoMT process [125]. Ricoguat, a sGC is an approved therapeutic agent for SSc PAH and has been found to be beneficial in Raynaud’s phenomenon [125]. 

### 10.4. Endothelin-1 Receptors Inhibitors

Treatment of SSc PAH patients with bosentan led to reduction in the higher serum level of serum soluble PECAM-1, ICAM-1, VCAM-1, and P-selectin after 12 months of therapy, suggesting attenuation of endothelial cell activation [126]. It also reduced higher levels of CD3-LFA1 T cell, suggestive of restoring T-cell function. In an invitro EndoMT model, microvascular endothelial cells (MVECs), when incubated with bosentan and macitentan, had reduced expression of mesenchymal markers and restoration of CD31 expression and the imbalance between VEGF-A and VEGF-A165b [127]. This suggests the endothelin-1 receptor inhibitors prevent EndoMT. These agents have been effective in treatment of SSc associated PAH and preventing development of digital ulcers [123].

### 10.5. Prostacyclins

Vascular endothelial cells synthesize prostacyclins, which possess a vasodilatory effect and inhibit platelet aggregation and vascular smooth muscle proliferation [128]. Various prostacyclin analogs (epoprostenol, trepostinil, iloprost) and the prostacyclin receptor inhibitor, selixipag, promote vasodilation through induction of the prostacyclin pathway. Prostacyclin analogs have been shown to have beneficial effects in the treatment of Raynaud phenomenon, digital ulcers, and PAH in SSc subjects [123]. Tsou et al. showed IV iloprost has a long-lasting effect for weeks after infusion of iloprost, which has a short half-life otherwise [129]. Following the infusion of iloprost, there were increased endothelial cell junctional VE-cadherin clustering, VE-cadherin level, tubulogenesis and inhibition of monolayers permeability, and EndoMT. The investigators suggested these endothelial effects may account for the long-lasting effect of IV iloprost following discontinuation of infusion. 

### 10.6. Cyclophosphamide

Cyclophosphamide has been widely used to treat SSc-associated interstitial lung disease. It also has beneficial effect on peripheral microvasculature [130]. The investigators showed treatment with cyclophosphamide showed better proliferation of dermal microvascular endothelial cells (MVECs) and less apoptosis. This might be related to a reduction in pretreatment levels of antiangiogenic factors such as pentraxin-3, MMP-12, angiostatin, and endostatin. Cyclophosphamide increases the number of circulating endothelial progenitor cell numbers and reduces serum VEGF levels, E-selectin and thrombomodulin, suggestive of improvement of endothelial cell damage [131,132,133].

### 10.7. Statins

Statins have been studied in the treatment of SSc. In a small study of 14 patients with SSc using atorvastatin 10 mg/day for 12 weeks, the number of bone marrow–derived circulating endothelial precursors (CEPs) increased, and improvement of Raynaud symptoms occurred [134]. In another study of SSc patients using simvastatin 20 mg/day over 12 weeks, there was no increase in endothelial progenitor cells but there was a reduction in the number of circulating endothelial cells [135]. There was also a reduction in levels of endothelin-1, soluble E-selectin, intercellular adhesion molecule-1, vascular cell adhesion molecule-1, and interleukin-6, suggesting that statin inhibits endothelial cell activation. 

### 10.8. Nitrate Therapy

Nitrates are metabolized into NO and they increase c-GMP concentration in vascular smooth muscle, leading to vasodilation [136]. Nitroglycerine tapes, MQX-503 (a novel nitroglycerine compound), and topical glyceryl trinitrate have been tried to improve NO levels in the peripheral circulation of patients with SSc suffering from Raynaud phenomenon and digital ulcers but met with variable levels of success [137,138,139].

### 10.9. Stem Cell Therapy

Stromal vascular fraction (SVF), isolated from the adipose tissue, contains mesenchymal stem cells, endothelial precursor cells, and T regulatory cells, etc. [140]. The mesenchymal stem cells, otherwise known as adipose-derived stem cells, can be further isolated from SVF. Local injection of these stem cells leads to the secretion of VEGF and fibroblast growth factor, which promote local angiogenesis [141], inhibit apoptosis, and promote endothelial cell proliferation [142]. A recent meta-analysis of the effectiveness of the adipose-derived stem cell injections showed improvement in digital ulcers, Raynaud symptoms, and an increase in the number of nail fold capillaries [143]. Autologous hematopoietic stem cell transplantation has a limited impact on vasculature. It only partially restored microvascular architecture [144] but had no effect on dermal vessel density [145]. Mesenchymal stromal cells (MSCs), isolated from adipose tissue or bone marrow, possess immunosuppressive, proangiogenic, and antifibrotic potential [146]. Allogenic and autologous IV infusion of MSCs have led to improvement in digital ulcers and peripheral circulation but the efficacy is limited to case reports and case series at the present time. 

## 11. Conclusions

We have reviewed the critical role of endothelial cells in the pathogenesis of SSc. Endothelial cell injury is the initial event in scleroderma. Besides endothelial injury, defective angiogenesis and vasculogenesis involving the endothelial cells play key roles. Furthermore, endothelial to mesenchymal cell transformation leads to tissue fibrosis. Therapeutic advances targeting the various aspects of endothelial dysfunction would be useful. 

## Figures and Tables

**Figure 1 ijms-24-14385-f001:**
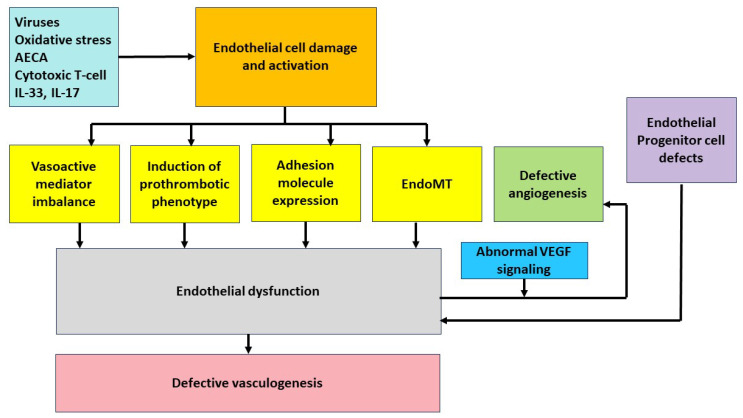
Endothelial dysfunction in systemic sclerosis; AECA: anti-endothelial cell antibodies, VEGF: vascular endothelial growth factor, EndoMT: endothelial to mesenchymal transition.

**Table 1 ijms-24-14385-t001:** Mediators of Endothelial to Mesenchymal Transition (EndoMT) [74].

Mediators	Mode of Action
TGF-β	Smad-dependent and independent pathways, e.g., c-abl kinase, protein kinase c-δ
Caveolin-1(CAV1)	Modulation of TGF-β signaling
Endothelin-1 (ET-1)	Synergic effect with TGF β, involving Smad pathway
Notch pathway	Activation of Snail and upregulate Smad
Wnt pathway	Smad-dependent autocrine TGF-β signaling
Hypoxia-inducible factor-1α (HIF-1α)	Interaction with TGF-β and VEGF, activation of Snail

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
