# Peer review of "Endothelial Dysfunction in Systemic Sclerosis"

_ijms, 2023, doi:10.3390/ijms241814385_

Round 1

Reviewer 1 Report

This is a well-written review paper featuring comprehensive topics relevant to endothelial dysfunction in SSc. This reviewer has only minor comments and suggestions, as shown below:

The term “systemic sclerosis (SSc)” should be consistently used throughout the manuscript. It is better to avoid use of “scleroderma” because this term contains not only SSc but also other diseases/conditions with thickened skin lesions such as morphia, and is often used as the term for lay persons. 

Vascular pathogenesis in SSc patients consists of at least two different mechanisms, including vasospasm and morphologic abnormality. This is important to understand clinical presentation and pathogenesis of SSc vasculopathy. 

The term “localized and systemic scleroderma” is incorrect, and must be changed to “limited and diffuse cutaneous SSc”.

“Anti-topoisomerase antibodies” should be corrected to “anti-topoisomerase I antibodies”.

Reference #86 should be removed because experiments using Klf5+/-; Fli1+/- mice have never replicated. 

Page 7: Nailfold capillaroscopy assesses morphology, and is unable to evaluate endothelial function. 

Soluble guanylate cyclase stimulators/activators should be included in the therapeutic considerations section. 

Page 8: In terms of cyclophosphamide, an important reference is missing (doi: 10.1093/rheumatology/keq259. PMID: 20724431). 

Author Response

  Thank you very much for taking the time to review the manuscript. Please find the detailed responses below in red and the corresponding revisions/corrections highlighted/in track changes in the resubmitted files     Comments and Suggestions for Authors

This is a well-written review paper featuring comprehensive topics relevant to endothelial dysfunction in SSc. This reviewer has only minor comments and suggestions, as shown below:

The term “systemic sclerosis (SSc)” should be consistently used throughout the manuscript. It is better to avoid the use of “scleroderma” because this term contains not only SSc but also other diseases/conditions with thickened skin lesions such as morphia, and is often used as the term for laypersons. Thank you. we have changed this to reflect systemic sclerosis (SSc) throughout the manuscript. 

Vascular pathogenesis in SSc patients consists of at least two different mechanisms, including vasospasm and morphologic abnormality. This is important to understand the clinical presentation and pathogenesis of SSc vasculopathy. Thank you for the suggestion. We agree and this is reflected in the introduction section. 

The term “localized and systemic scleroderma” is incorrect, and must be changed to “limited and diffuse cutaneous SSc”. Thank you. This has been corrected in the introduction section. 

“Anti-topoisomerase antibodies” should be corrected to “anti-topoisomerase I antibodies”. Thank you. We have corrected it. 

Reference #86 should be removed because experiments using Klf5+/-; Fli1+/- mice have never replicated. Thank you. we have deleted the reference and relevant section from the text

Page 7: Nailfold capillaroscopy assesses morphology, and is unable to evaluate endothelial function. Thank you. we have changed the assessment section to morphological assessment for nail fold capillaroscopy and other methods for functional assessment

Soluble guanylate cyclase stimulators/activators should be included in the therapeutic considerations section. Thank you. we have added a paragraph reflecting this in the therapeutic section. 

Page 8: In terms of cyclophosphamide, an important reference is missing (doi: 10.1093/rheumatology/keq259. PMID: 20724431). Thank you. we have added that reference to the therapeutic section under cyclophosphamide. 

Reviewer 2 Report

In this narrative review, Patnaik and coworkers outlined the mechanisms of endothelial dysfunction in SSc, underlining the importance of this disruption in the pathogenesis of the diseases.

Overall, the review is well written and comprehensive, although it could be better organized.

For example, I would unify the paragraphs regarding pathogenic mechanisms into a single paragraph and then use sub-headings; and then continue the review of clinical aspects, assessment and therapeutic measures. It would also be helpful to the reader for the Figure to match the subheadings of the pathogenesis paragraph, as this could serve as a "take-home message".

Minor comments

I would remove the ":" after the title of the paragraph.

Lines 26-27. This sentence appears to have no verb.

Table 1 is fine but should be inserted in the respective paragraph. Perhaps the authors could add appropriate references in the Table.

English language is fine but minor spell checks are needed.

Author Response

Thank you very much for taking the time to review this manuscript. Please find the detailed responses below in red text and the corresponding revisions/corrections highlighted/track changes in the resubmitted files.    Comments and Suggestions for Authors

In this narrative review, Patnaik and coworkers outlined the mechanisms of endothelial dysfunction in SSc, underlining the importance of this disruption in the pathogenesis of the diseases.

Overall, the review is well written and comprehensive, although it could be better organized. Thank you

For example, I would unify the paragraphs regarding pathogenic mechanisms into a single paragraph and then use sub-headings; and then continue the review of clinical aspects, assessment and therapeutic measures. It would also be helpful to the reader for the Figure to match the subheadings of the pathogenesis paragraph, as this could serve as a "take-home message". We agree and have merged it into a single paragraph and used the subheadings as suggested. We have referred to the figure in the appropriate section. 

Minor comments

I would remove the ":" after the title of the paragraph. This is corrected

Lines 26-27. This sentence appears to have no verb. This is corrected. 

Table 1 is fine but should be inserted in the respective paragraph. Perhaps the authors could add appropriate references in the Table. We have inserted the table in the appropriate section and we added the reference. 

Comments on the Quality of English Language

English language is fine but minor spell checks are needed. : We have corrected the spelling.